# Contact pressure explains half of the abdominal aortic aneurysms wall thickness inter-study variability

**Jan Kracík[1], Luboš Kubíček[2], Robert Staffa[2], Stanislav Polzer[3]***

**1** Department of Applied Mathematics, VSB-Technical University of Ostrava, Ostrava, Czech Republic, **2** 2nd Department of Surgery, St. Anne's University Hospital Brno and Faculty of Medicine, Masaryk University, Brno, Czech Republic, **3** Department of Applied Mechanics, VSB-Technical University of Ostrava, Ostrava, Czech Republic

* Polzer@seznam.cz

**Data Availability Statement:** All relevant data are within the manuscript and its Supporting information files. The code where Bayesian model is applied to generate distribution functions of the individual model parameters are held in publis

## Abstract

The stochastic rupture risk assessment of an abdominal aortic aneurysm (AAA) critically depends on sufficient data set size that would allow for the proper distribution estimate. However, in most published cases, the data sets comprise no more than 100 samples, which is deemed insufficient to describe the tails of AAA wall thickness distribution correctly. In this study, we propose a stochastic Bayesian model to merge thickness data from various groups. The thickness data adapted from the literature were supplemented by additional data from 81 patients. The wall thickness was measured at two different contact pressures for 34 cases, which allowed us to estimate the radial stiffness. Herein, the proposed stochastic model is formulated to predict the undeformed wall thickness. Furthermore, the model is able to handle data published solely as summary statistics. After accounting for the different contact pressures, the differences in the medians reported by individual groups decreased by 45%. Combined data can be fitted with a lognormal distribution with parameters $\mu = 0.85$ and $\sigma = 0.32$ which can be further used in stochastic analyses.

## 1 Introduction

An abdominal aortic aneurysm (AAA) is a continuously growing dilatation of the abdominal aorta. The main danger lies in possible rupture, which is a life-threatening situation with a mortality rate of about 50% [1], which remains unchanged over time even though substantial effort to predict AAA rupture has been made. On the other hand, not all AAAs have the tendency to rupture and surgical intervention may be also associated with non-negligible risks and a possibility of reduced quality of life afterward. Therefore, selectively intervening only in cases with a significant risk of rupture is extremely important. So far, this selection has been based on the maximal diameter of the AAA and its growth rate [2]. The main limitation of the current threshold approach (a maximal diameter of 55 mm for men and a growth of 10 mm per year) is that it is population-based. As the methods do not account for individual patient data, the accuracy is only limited. It has been shown that the probability of a rupture of the

repository: Kracík, Jan; Polzer, Stanislav (2024), "Contact pressure explains half of the abdominal aortic aneurysms wall thickness inter-study variability", Mendeley Data, V1, doi: 10.17632/pnfmmw8m89.1 The other parts of computations were done interactively using various tools and commercial software and cannot be put into single code which was not our aim in this paper. We are prepared to share this detailed step by step process upon request send to email jan.kracik@vsb.cz.

**Funding:** This work was supported by Specific Research „Experimental and Numerical Modeling for Solving Problems in Mechanics and Biomechanics"(SP2024/037) awarded to SP.

**Competing interests:** The authors have declared that no competing interests exist.

AAA with a diameter smaller than 7 cm is only 10% per year [3], meaning that most of the interventions are performed on stable AAAs, which would not rupture in the next year. Therefore, researchers aim to identify additional criteria that could improve the rupture prediction accuracy. Among the suitable candidates, the possibly most promising are based on wall stress [4, 5] which is typically assessed via finite element analysis [6–8]. The current effort is to increase the accuracy of the computed wall stress which depends on multiple factors that are known only with a significant uncertainty even on a patient-specific basis [9].

The wall thickness is an exemplary case. It has been shown that wall thickness in a normal abdominal aorta depends on sex, race, and age [10], but similar observations have not been confirmed so far in the case of an AAA [11, 12]. Instead, the wall thickness in an AAA was shown to be thinner in the posterior compared to the anterior [13] and thinner behind the intraluminal thrombus [14]. Conclusions regarding the wall thickness in intact vs. ruptured AAAs are conflicting. One study reported that ruptured AAAs have thicker walls [15] but others failed to confirm that [16, 17]. Table 1 shows the reported values of AAA wall thickness. The variations between the studies are significant, which attributed to various measurement methods [18].

Wall thickness is further known for having large inter [9] and intra [19] patient variability, yet the wall stress critically depends on it. Therefore, there is considerable uncertainty in computational AAA rupture risk predictions [8, 20] because of the unattainability of the patient-specific wall thickness directly from computer tomography or magnetic resonance images due to the presence of other structures with a similar density and appearance (e.g., fatty tissue, the inferior vena cava, or the intraluminal thrombus). Thus, it is critical to estimate the likelihood of low wall thickness occurrence where a rupture is eventually located [19] despite a known increase in the wall strength of thinner samples [21–23]. However, to correctly estimate the likelihood of low wall thickness would require extensive data sets that no single study can provide due to the rare occurrence of the phenomena. The most straightforward solution would be combining data from various studies, but the described systematic shift precludes such a direct approach. Moreover, some of the studies only report the mean and standard deviation of measured thicknesses instead of complete data sets [15, 22] That further complicates the linking of such data together.

**Table 1. A list of studies used to generate the wall thickness distribution via the probabilistic model together with evaluated mean wall thickness and mean undeformed wall thickness using the proposed model.**

| Study | Number of AAAs | Number of samples | Thickness measurement method | Contact pressure [kPa] | Mean/median wall thickness from the study [mm] | Mean/median wall after accounting for contact pressure (study results) [mm] | Note |
|---|---|---|---|---|---|---|---|
| Di Martino et al. [15] | 25 | 39 | Laser micrometer | 0 | 2.25/2.02 | 2.25/2.02 | |
| Martufi et al. 2015 [22] | 27 | 34 | Laser micrometer | 0 | 2.71/2.59 | 2.71/2.59 | results only in the form of mean ±SD |
| Bruder et al. [20] | 113 | 251 | Micrometer 0.5N | 25kPa | 1.71/1.62 | 2.55/2.42 | |
| Tong et al. [11] | 90 | 90 | Laser micrometer | 0 | 2.56/2.5 | 2.56/2.5 | results only in the form of mean ±SD |
| Polzer et al.–uniaxial [23] | 45 | 90 | Thickness gauge | 16kPa | 1.49/1.55 | 2.16/2.19 | patients (not samples) partially overlap with the current study |
| Polzer et al.–biaxial–current study | 75 | 92 | Thickness gauge | 1kPa | 1.92/1.69 | 2.24/2.22 | patients (not samples) partially overlap with (23) |
| Difference in ($mean_{max} - mean_{min}$)/in ($median_{max} - median_{min}$ [mm] | | | | | **1.22/1.04** | **0.55/0.57** | |

Several studies have tried to respect variability in wall thickness. Some use the neural network [24, 25] while others apply the Bayesian approach [26, 27]. However, both approaches have a significant drawback: they depend on the underlying data, which are not publicly available and without them these approaches are rendered unviable.

This motivated us to propose a Bayesian stochastic model that allows combining data from various studies, including those that only present statistics instead of actual measurement data. Subsequently, the presented approach allows for a more precise estimate of a very low wall thickness (below 1mm) for which any single available data set is too small.

## 2 Methods

As an initial step, we gathered the aneurysmal wall thickness measurements under various contact pressures to estimate the relation between contact pressure and the measured wall thickness and subsequently evaluate the undeformed wall thickness. A reliable calculation of undeformed wall thickness requires information about the radial stiffness of the sample which, to the best of our knowledge, has not yet been presented.

### 2.1 Sample acquisition

Patients included in this study underwent an open surgical repair of their asymptomatic AAA at St. Anne's University Hospital in Brno between the 8th of February 2011 and the 29th of April 2016. The surgeons removed a part of the aneurysm tissue from the anterior part of the aneurysmal sac, approximately in the middle between renal arteries and iliac bifurcation. All procedures were approved by the St. Anne's University Hospital Ethics Committee (49V/ 2010), all the patients signed an informed consent on the procedure. The following patient-specific data were gathered: age, sex, AAA diameter (Dmax), hypertension, mean arterial pressure (MAP), chronic obstructive pulmonary disease (COPD), and hyperlipidemia (HLP). This patient-specific information can be found in the supplemental material. A total of 86 patients (out of which 19 were females) was included in the study.

The freshly harvested tissue was inserted into an insulation box with pre-heated 0.1% saline solution and transported immediately to the lab for mechanical testing. The testing procedure followed the steps as described in detail elsewhere [23]. Briefly, dog-bone (the narrow part 2mm, a total length of 40mm) and square-shaped (18x18mm) specimens were cut out from the adjacent regions of the sample for uniaxial and biaxial testing, respectively. Specimen dimensions were given by the cutter dimensions and are related to the dimensions during cutting. The specimen was then placed between two pieces of glass and its thickness was measured using a thickness gauge under a constant weight (a thickness gauge spindle + top glass) of 3m3g, which resulted in a contact pressure of $p_{c\_uni} = 16\ kPa$ for small uniaxial specimens (further referred to as uniaxial as in Polzer et al. [23]) and $p_{c\_biax} = 1$ kPa for much larger biaxial specimens (further referred to as biaxial as in Polzer et al.). Note that the data from uniaxial testing were already published [23]. Herein, we combine it with data from larger specimens not used in the previous study, resulting in a data set of 182 samples. All measurements with patient-specific information can be found in S1 File. We acquired wall thickness information from both dog-bone and square-shaped samples from 34 patients. These samples thus came from adjacent regions and as a result, the data contain systematic shifts in wall thickness measurements caused by various contact pressures. Additionally, we were able to prepare more than one sample from the harvested tissue in 53 cases. This was further exploited in training the hierarchical Bayesian model proposed in Section 2.3 to account for both the effect of various contact pressures and intra-patient variability.

## 2.2 Undeformed and deformed wall thickness prediction

Radial stiffness can be estimated via finite elasticity theory if the sample thickness is measured under at least two known radial pressures $p$. Here, we assume that the deformation of aneurysmal specimens in the radial direction can be described via the standard Holzapfel Gasser Ogden model [28] with the strain energy density function $\Psi$ consisting of isotropic $\Psi_{iso}$ and anisotropic $\Psi_{aniso}$ part as follows:

$$\boldsymbol{\Psi} = \boldsymbol{\Psi}_{iso} + \boldsymbol{\Psi}_{aniso} = \frac{c}{2} \cdot (I_1 - 3) + \frac{c}{2 \cdot k_2} \left( e^{\left(k_2(I_4-1)^2\right)} - 1 + e^{\left(k_2(I_6-1)^2\right)} - 1 \right), \quad (1)$$

Where $I_1$ denotes the first invariant of the left Cauchy-Green deformation tensor **B**, $c$, $k$ [MPa] and $k_2$ stands for fiber initial stiffness and level of stiffening, respectively. Further, invariants $I_4 = \boldsymbol{a_{01}} \cdot \boldsymbol{Ca_{01}}$ and $I_6 = \boldsymbol{a_{02}} \cdot \boldsymbol{Ca_{02}}$ are related to the directions of the two families of fibers with respect to the circumferential direction. **C** stands for the right Cauchy-Green deformation tensor. Direction vectors in an undeformed state hold $\boldsymbol{a_{01}} = (cos\varphi, sin\varphi, 0)^T$ and $\boldsymbol{a_{02}} = (-cos\varphi, sin\varphi, 0)^T$ with $\varphi$ referencing a declination of each family of fibers from the circumferential direction in radians. The deviatoric part of **B** for the uniaxial compression of anisotropic volumetrically incompressible material follows:

$$\bar{\boldsymbol{b}} = \begin{bmatrix} \dfrac{\lambda_c^2}{J^{2/3}} & 0 & 0 \\[2ex] 0 & \dfrac{\lambda_a^2}{J^{2/3}} & 0 \\[2ex] 0 & 0 & \dfrac{\lambda_r^2}{J^{2/3}} \end{bmatrix}, \quad (2)$$

where $\lambda_c$, $\lambda_a$, $\lambda_r$ represent the circumferential, axial and radial stretch respectively and $J$ stands for Jacobian of deformation gradient tensor $\boldsymbol{F}$. The first Piola Kirchhoff (FPK) stress tensor $\boldsymbol{P}$ can be derived from Eq (1) as derived elsewhere (28):

$$\boldsymbol{P} = \boldsymbol{F}^{-1} \left( -p_h \cdot \boldsymbol{I} + 2 \frac{\partial W_{iso}}{\partial \overline{I}_1} \cdot dev\,\bar{\boldsymbol{b}} + 2 \left( \frac{\partial \Psi_{aniso}}{\partial \overline{I}_4} \cdot dev(\boldsymbol{a}_1 \otimes \boldsymbol{a}_1) + \frac{\partial \Psi_{aniso}}{\partial \overline{I}_6} \cdot dev(\boldsymbol{a}_2 \otimes \boldsymbol{a}_2) \right) \right) \quad (3)$$

Where $p_h$ represents hydrostatic pressure, $\boldsymbol{a_1} = \overline{\boldsymbol{F}}\boldsymbol{a_{01}}$ and $\boldsymbol{a_2} = \overline{\boldsymbol{F}}\boldsymbol{a_{02}}$ represent a deformed direction vector, $\overline{\boldsymbol{F}} = (J^{-1/3}\boldsymbol{I})\boldsymbol{F}$ is the deviatoric part of the deformation gradient tensor. The radial stress cannot be derived explicitly from Eq (3), so we solved it numerically. The number of variables needs to be reduced as only two experimental points on the pressure-deformation curve are available. To do this, the mentioned constitutive model was used to simultaneously fit all available biaxial responses of the aneurysmal tissue published in [29] to obtain a mean response. Such fitting resulted in the following constants: $\varphi = 0.546$, $k_2 = 15$. Constants $c$, $k$ were kept as variables because they govern the response in the low radial compression region where our measurement data points are located. FPK radial stress $P_r$ (which equals contact pressure $p_r$) was evaluated on a regular grid for $c \in \langle 0,10 \rangle$ $kPa$, $k \in \langle 0,40 \rangle$ $kPa$, and the radial stretch in the range $\lambda_r \in \langle 0.61,1 \rangle$ in order to cover the expected range of radial deformations in the specimen. The resulting matrix of 4 x 34 440 points was used to fit the response function as follows. For each pair of $c$ and $k$ values the dependency of radial stress $P_r$ on the deformation $\lambda_r$ was approximated with a cubic spline, and deformation values which led to given radial pressure values $p_r$ (as applied to the samples during the measurement procedure) were evaluated numerically. For each of the pressure values, the response of radial deformation $\lambda_r$ with relation to material constants $c$ and $k$ was fitted with a non-uniform seventh-order spline as a

function $\lambda_r(c, k, p_r)$. This function is further used within a hierarchical Bayesian model (see Section 2.3), the parameters of which are inferred via a Markov chain Monte Carlo algorithm based on a no U-turn sampler [30]. A sufficiently smooth and tractable response function is essential for converging the inference algorithm. The deformed thickness $t_{ij}$ of the $j$-th sample from the $i$-th patient is then expressed as

$$t_{ij} = \lambda_{r\_ij} T_{ij}, \tag{4}$$

where $T_{ij}$ denotes the undeformed wall thickness of the $i$-th patient and $j$-th sample and $\lambda_{r\_ij}$ enotes its radial stretch. Due to the low number of available data, the material parameters $c,k$ are kept constant for all cases (both female and male patients). These constants are estimated using the Bayesian model as described further.

## 2.3 Hierarchical Bayesian model

The data analysis is based on a hierarchical probabilistic model which combines the information from six data sets summarized in Table 1. Due to a significant number of cases where more than one sample was available from one patient, the wall thickness variability is decomposed into inter-patient and intra-patient parts. This allows us to account for different overall wall thicknesses between individual patients as well as variations in thickness between samples from the same patient [19]. Without this correction, the estimates of model parameters would be biased toward the mean wall thickness of patients with higher sample numbers.

The undeformed thicknesses $T_{ij}$ of the $i$-th patient is assumed to have the lognormal distribution $LN(\mu_i, \sigma_{pat})$. Note that the lognormal distribution is commonly used in the literature as a probabilistic model of wall thickness [8, 20]. The parameters $\mu_i$ are patient-specific, while the parameter $\sigma_{pat}$ is retained for all the patients and quantifies the intra-patient variability. Consequently, individual patients have different wall thickness distribution. The parameters $\mu_i$ are modeled as independent, identically distributed random variables with the Gaussian distribution $N(\mu_{pop}, \sigma_{pop})$, where the parameter $\sigma_{pop}$ quantifies the inter-patient variability. Under these assumptions, the probability distribution of the wall thickness of an arbitrary sample is $LN(\mu_{pop}, \left(\sigma_{pop}^2 + \sigma_{pat}^2\right)^{0.5})$. The undeformed wall thicknesses $T_{ij}$ are then transformed to deformed thicknesses $t_{ij}$ by the Eq (4) where $\lambda_{r\_ij}$ depends on the unknown parameters $c$, $k$, and known pressure values $p_{ij}$. Note that the samples from studies where a non-contact thickness measurement was used [15, 22] can be equivalently taken as samples deformed by a pressure of 0 kPa. Altogether, our model contains five unknown parameters $c$, $k$, $\mu_{pop}$, $\sigma_{pop}$ and $\sigma_{pat}$ to describe the distribution of wall thickness. Fig 1 shows the visualization of the model in the form of a graphical model [31], including the summary statistics discussed in Section 2.3.1.

A standard Bayesian approach [32] is employed to estimate the model's unknown parameters. Due to the lack of prior information about the unknown parameters, we use independent uniform prior distributions on sufficiently wide intervals for the individual parameters, as depicted in Fig 3. To assess the robustness of the model with respect to the choice of the prior distribution, the analysis was also performed with four alternative prior distributions. For details, see S2 File. The posterior distribution is approximated by the Markov Chain Monte Carlo method using a Python library PyMC3 [33]. Altogether 200 000 samples are sampled from 20 independent Markov chains (10 000 samples per chain). The convergence is checked with the R-hat diagnostic test [34]. The effective sample sizes are between 108 000 and 282 000 samples.

**2.3.1 Sub-Model to account for the data presented as summary statistics.** The above-described model is applicable for measurement data sets, however, several studies refrain from

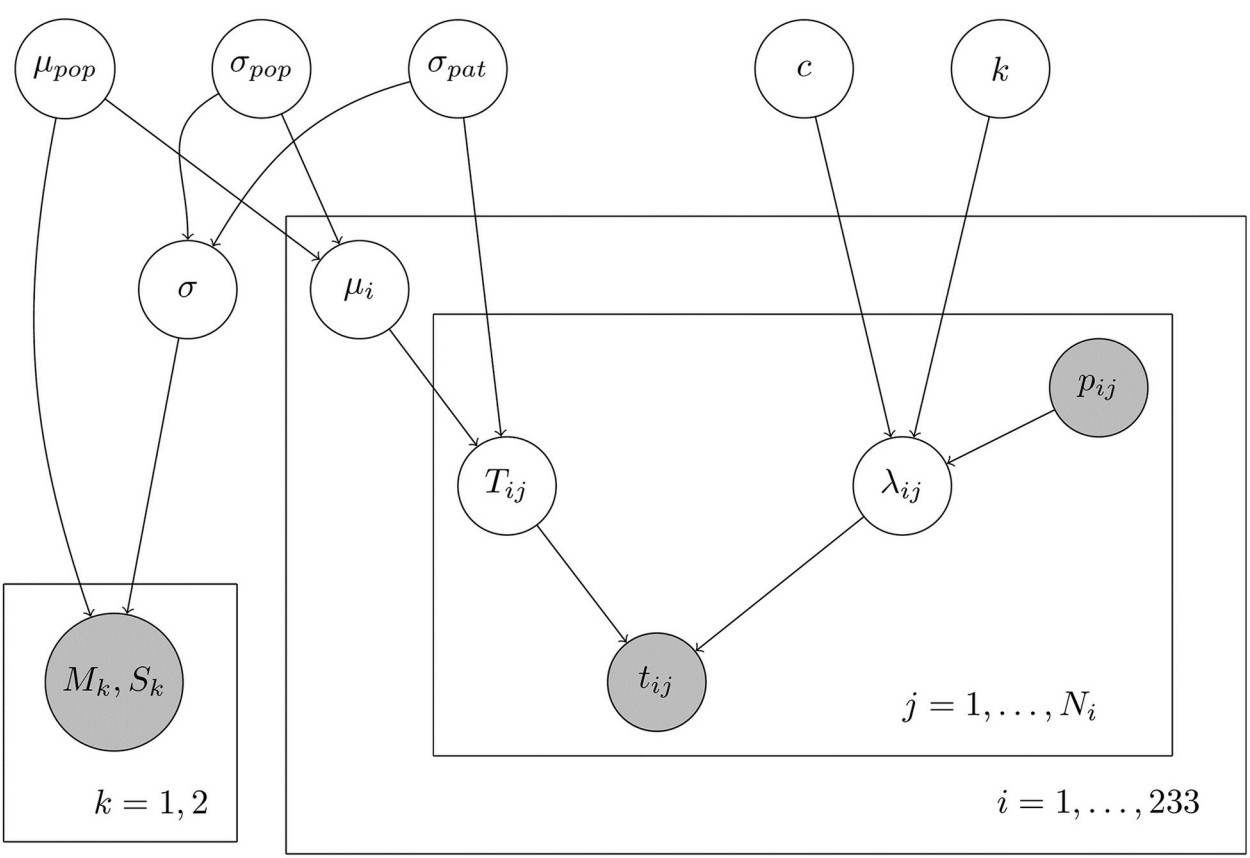

$$\mu_{pop} \sim U(0.4, 1.4) \qquad \mu_i \sim \mathcal{N}(\mu_{pop}, \sigma_{pop}) \qquad \sigma = \left(\sigma_{pop}^2 + \sigma_{pat}^2\right)^{0.5}$$
$$\sigma_{pop} \sim U(0.05, 0.4) \qquad T_{ij} \sim \mathcal{LN}(\mu_i, \sigma_{pat}) \qquad M_k, S_k \sim f_k(M_k, S_k; \mu_{pop}, \sigma)$$
$$\sigma_{pat} \sim U(0.05, 0.4) \qquad \lambda_{r\_ij} = \lambda_r(c, k, p_{ij})$$
$$c \sim U(0, 10) \qquad\qquad t_{ij} = T_{ij}\lambda_{r\_ij}$$
$$k \sim U(0, 40)$$

**Fig 1. Graphical representation of the probabilistic model.** Graph vertices filled in grey correspond to variables with known values. The groups of variables bounded in a rectangle repeat in the model. Individual repetitions are marked with an index in the bottom right corner of the rectangle.

publishing the complete data sets and report their measurements as summary statistics only [11, 22]. These data also contain valuable information, but their combination with whole measurement data sets is not straightforward. Let us consider studies that report summary statistics in the form of sample mean and standard deviation which will be further denoted as $M_k$, $S_k$ respectively with the subscript $k$ referring to the individual study. The statistics $M_k$, $S_k$, being deterministic functions of raw data, are random variables, the distribution of which is given by the values of parameters $\mu_{pop}$, $\sigma_{pop}$, $\sigma_{pat}$ and the sample size. Recall that these data sets contain undeformed thicknesses, and as such, they do not depend on parameters $c$ and $k$. Since there is no known closed-form formula or tractable approximation of the distribution of the sample mean and the standard deviation of independent lognormally distributed variables, a computationally intensive approximation of the joint density of the sample mean and standard

deviation based on the Fourier transform had to be employed. The statistics $(M_k, S_k)$ are one-to-one functions of random vectors $V_k = \left( \sum_{i=1}^{N_k} T_{k;i}, \sum_{i=1}^{N_k} T_{k;i}^2 \right)$, where $T_{k;i}$ denotes the undeformed thickness of $i$-th sample from the $k$-th data set and $N_k$ is the number of samples in the $k$-th data set. To identify the distribution of $(M_k, S_k)$ it is thus sufficient to know the joint distributions of random vectors $V_k$ and apply the Change of Variables formula. Since $V_k$ is a sum of $N_k$ independent random vectors $(T_{k;i}, T_{k;i}^2)$, its distribution is the $N_k$-fold convolution of the distributions of $(T_{k;i}, T_{k;i}^2)$. We compute the convolution using the Fourier transform of the distribution of $(T_{k;i}, T_{k;i}^2)$. By the convolution theorem, the Fourier transform of the convolution of probability measures is a product of the Fourier transforms. The Fourier and the inverse Fourier transform are calculated using numerical integration methods. This way, the joint density $f_k(M_k, S_k; \mu_{pop}, \sigma)$ of $(M_k, S_k)$ can be evaluated for any pair of parameters $\mu_{pop}, \sigma$. For the two pairs of statistics $(M_k, S_k)$, the densities $f_k(M_k, S_k; \mu_{pop}, \sigma)$ as functions of $\mu_{pop}$ and $\sigma$ were evaluated on a grid of 100x100 points and fitted with bicubic splines.

## 2.4 Considered data sets

The proposed probabilistic model is applicable to any published data as long as the information about radial pressure is included. Therefore, all studies where wall thicknesses were measured by a caliper [16, 17, 35] had to be excluded from the study. Additionally, we excluded studies where wall thickness is estimated from the histological slices [12, 14] as the inherent chemical treatment of such samples alters the wall thickness significantly. As a result, only a limited amount of data sets is deemed admissible (listed in Table 1). Note that all of them comprise the samples harvested from the anterior part of the AAA, similar to our case. Two of the studies report aneurysmal wall thickness only in the form of summary statistics [11, 22]. While the data set Tong et al. [11] contains one sample per patient, the data set Martufi et al. [22] contains 34 samples from 27 patients. Because the number of patients for which there is more than one sample is negligible compared to the total number of patients in all of the included studies, we treated the 34 samples as independent samples from 34 patients. The distribution of each individual sample is then $LN(\mu_{pop}, \sigma)$, where $\sigma = \left( \sigma_{pop}^2 + \sigma_{pat}^2 \right)^{0.5}$. The values of the statistics are for Tong et al. and Martufi et al. $(M_1, S_1) = (2.71, 0.83)$ and $(M_2, S_2) = (2.56, 0.59)$, respectively. Note that in both of the papers Martufi et al. and Tong et al., the data set is split into two subsets (a thin/thick wall and male/female patients, respectively), and the statistics are reported for each subset separately. By combining the two statistics into one, we got a summary statistic for the complete data set.

Individual thickness measurements extracted from the studies in Table 1 were used as inputs for our model.

## 2.5 Model verification

The proposed model was verified in several ways. Firstly, the likelihoods $f(M_k, S_k; \mu_{pop}, \sigma)$ were plotted to verify that the model leads to the expected results of values $\mu_{pop}, \sigma$ concentrated around a single point for both statistics $M_1, S_1$ and $M_2, S_2$. The correctness of the calculation of the joint distribution of the statistics $M_k, S_k$ was also validated using Monte Carlo simulations. Next, by reviewing the distributions of each parameter, the suitability of the chosen ranges was verified. Furthermore, the joint distributions of stiffness-like parameters $c$ and $k$ and inter and intra-patient variability $\sigma_{pop}$ and $\sigma_{pat}$ revealed whether they are not highly dependent on each other. Finally, the predictive density of an undeformed thickness was compared with the

histogram of the undeformed thickness values, which consists of a mix of measured and estimated undeformed thicknesses.

## 2.6 Accounting for the systematic shift caused by contact pressure

Once the model was verified, we applied it to the data sets to estimate the undeformed aneurysmal thickness distribution for all considered studies indicated in Table 1. The data from studies using non-contact thickness measurements were not modified, while the data from studies using contact thickness measurements were corrected using the proposed model. The difference in extreme means ($mean_{max} - mean_{min}$) and medians ($median_{max} - median_{min}$) for the corrected and original data sets were chosen as metrics quantifying the systemic shifts caused by the contact thickness measurement.

Finally, we constructed a posterior predictive distribution from all included data sets and statistics while accounting for various contact pressures, fitted the result with a lognormal distribution, and estimated the probability of AAA wall thickness reaching very low values (below 1mm).

## 3 Results

### 3.1 Sub-model accounting for the data presented as summary statistics

The logarithms of densities $f_k(M_k, S_k; \mu_{pop}, \sigma)$ taken as functions of $\mu_{pop}$ and $\sigma$, called log-likelihoods, are depicted in Fig 2. The log-likelihoods are concentrated approximately around values ($\mu_{pop} = 0.95$, $\sigma = 0.3$) for study [22] and ($\mu_{pop} = 0.91$, $\sigma = 0.23$) for study [11] which are point estimates of the parameters of the lognormal distributions obtained by the Method of Moments from the statistics ($M_1$, $S_1$) and ($M_2$, $S_2$), respectively. Note that the log-likelihood based on the statistics from Tong et al. (right) is more concentrated than the log-likelihood based on the statistics from Martufi et al. (left) because the data set in Tong et al. is almost three times larger.

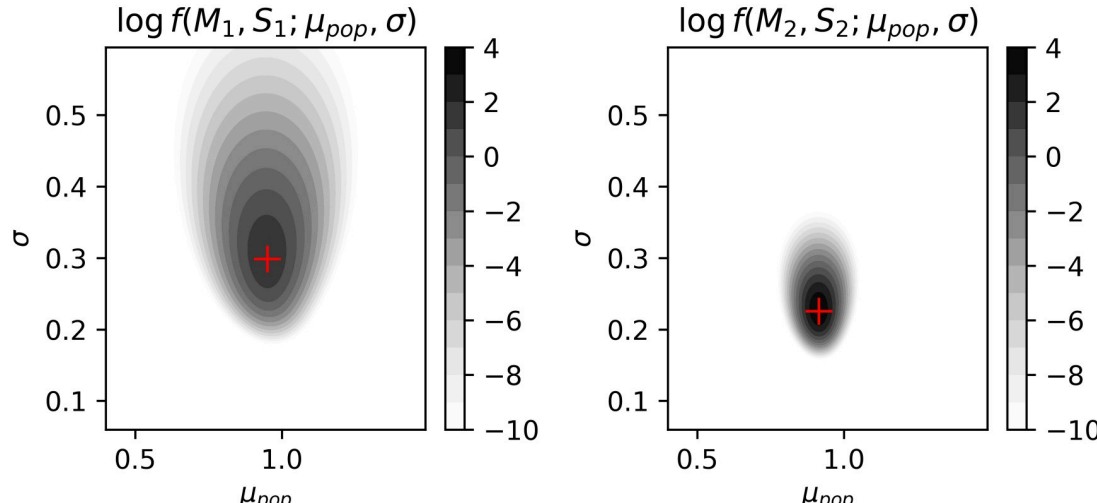

**Fig 2. The log-likelihoods of parameters $\mu_{pop}$ and $\sigma$ based on the statistics from Martufi et al. 2015 (left) and Thong et al. 2013(right).** The red plus signs correspond to point estimates of $\mu_{pop}$ and $\sigma$ obtained by the Method of Moments.

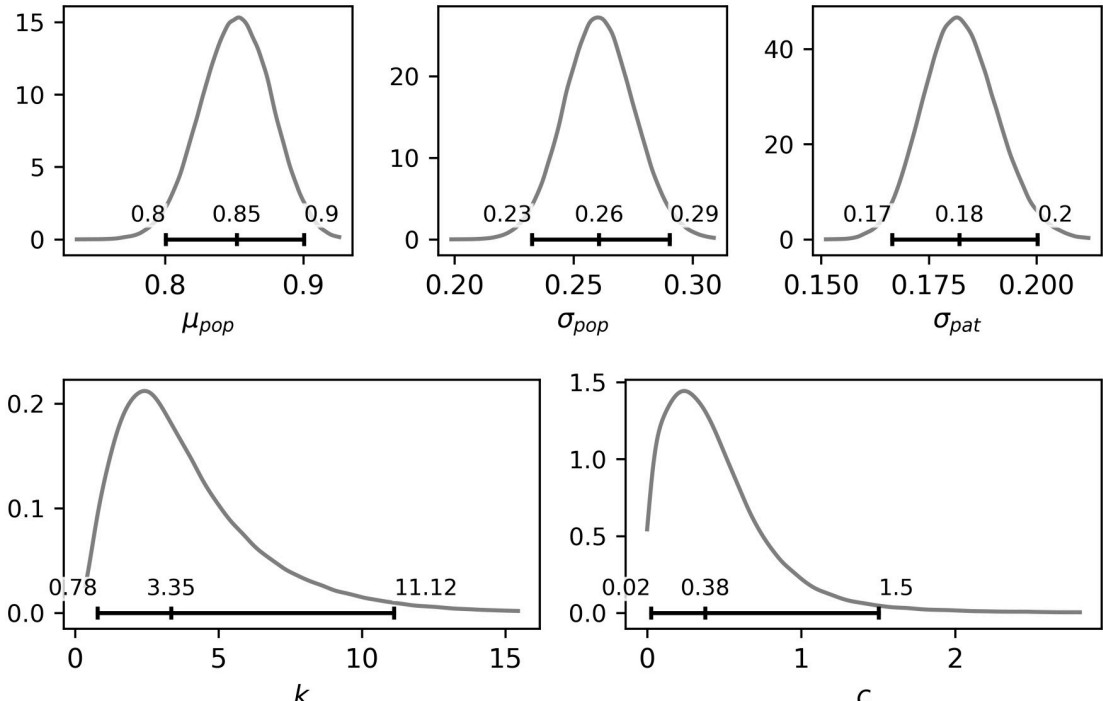

**Fig 3. The marginal posterior densities of parameters with 95% credible intervals and medians.**

## 3.2 Estimation of model parameters

The posterior densities of parameters together with 95% credible intervals (a Bayesian analogy of confidence intervals) and posterior medians are shown in Fig 3. The selected joint posterior densities of the pairs of parameters that could potentially exhibit strong mutual relationships are plotted in Fig 4. A strong relationship between $\sigma_{pop}$ and $\sigma_{pat}$ would occur if the data did not

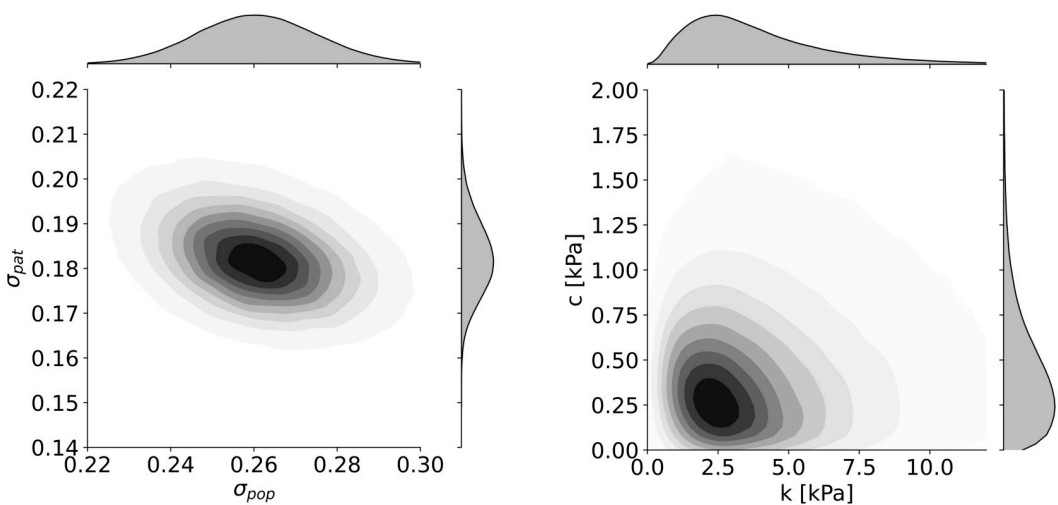

**Fig 4. The joint posterior densities of $\sigma_{pop}$ and $\sigma_{pat}$ (left) and $c$ and $k$ (right) show that the parameters do not exhibit a strong mutual relationship; therefore, they are all essential for the model definition.**

allow for distinguishing between the inter-patient and intra-patient variability. Similarly, the parameters $c$ and k would be strongly dependent if the similar values of deformations $\lambda_{r\_ij}$ could be achieved with different combinations of parameters $c$ and k. The joint posterior densities in Fig 4 indicate that there are no strong relations between the parameters.

**3.2.1 Inter-patient and intra-patient variability.** As expected, neither the inter-patient nor the intra-patient variability is negligible. The intra-patient variability represented by the parameter $\sigma_{pat}$ (posterior median 0.18) is by some 30% lower than inter-patient variability represented by the parameter $\sigma_{pap}$ (posterior median 0.26).

**3.2.2 Material parameters $c$ and $k$.** The combined data allow for inference on material parameters $c$ and $k$. Considering the relatively small number of data, the posterior densities are reasonably concentrated and do not exhibit a strong mutual relation (see the joint density in Fig 4), which supports the use of the two-parametric deformation model described in Section 2.2. From the joint posterior distribution of $c$ and $k$, the deformation $\lambda_r$ can be predicted as a function of pressure $p$. The median, 0.025 and 0.975 quantiles of deformation $\lambda_r$ depending on contact pressure p are plotted in Fig 5. The robustness analysis performed on four other prior distributions showed the posterior distributions of the parameters $c$ and $k$ were only as shown in S2 File.

## 3.3 Accounting for the systemic shift caused by contact pressure

Accounting for various contact pressures allowed for decreasing the discrepancy of the reported values between the groups, as shown in Fig 6. A comparison of data sets containing measured deformed thickness (boxplots 1–3), their transformations to undeformed thicknesses (estimated boxplots 4–6), and data sets with measured undeformed thicknesses (boxplots 7–9) are depicted in Fig 6. The boxplots for data sets Martufi et al. and Tong et al. are

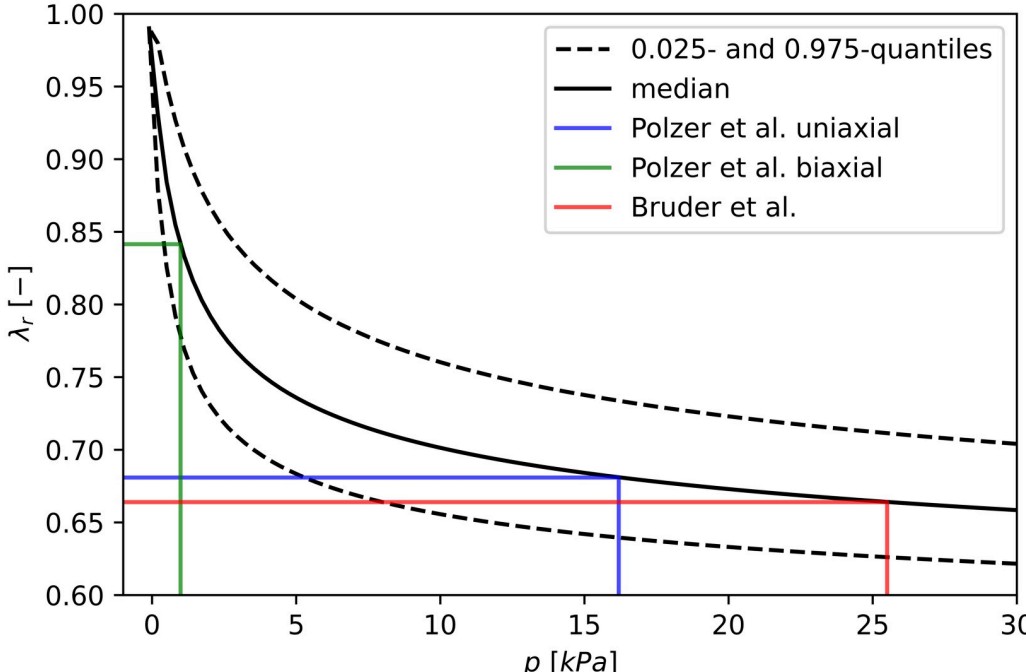

**Fig 5. The prediction of the radial stretch depending on contact pressure as a consequence of parameters kept fixed $\varphi = 0.546$, $k_2 = 15$ and estimated material parameters $c$, $k$ via HGO constitutive model.** The strong effect of contact pressure on the radial deformation explains the distinctly different mean thicknesses reported by various groups.

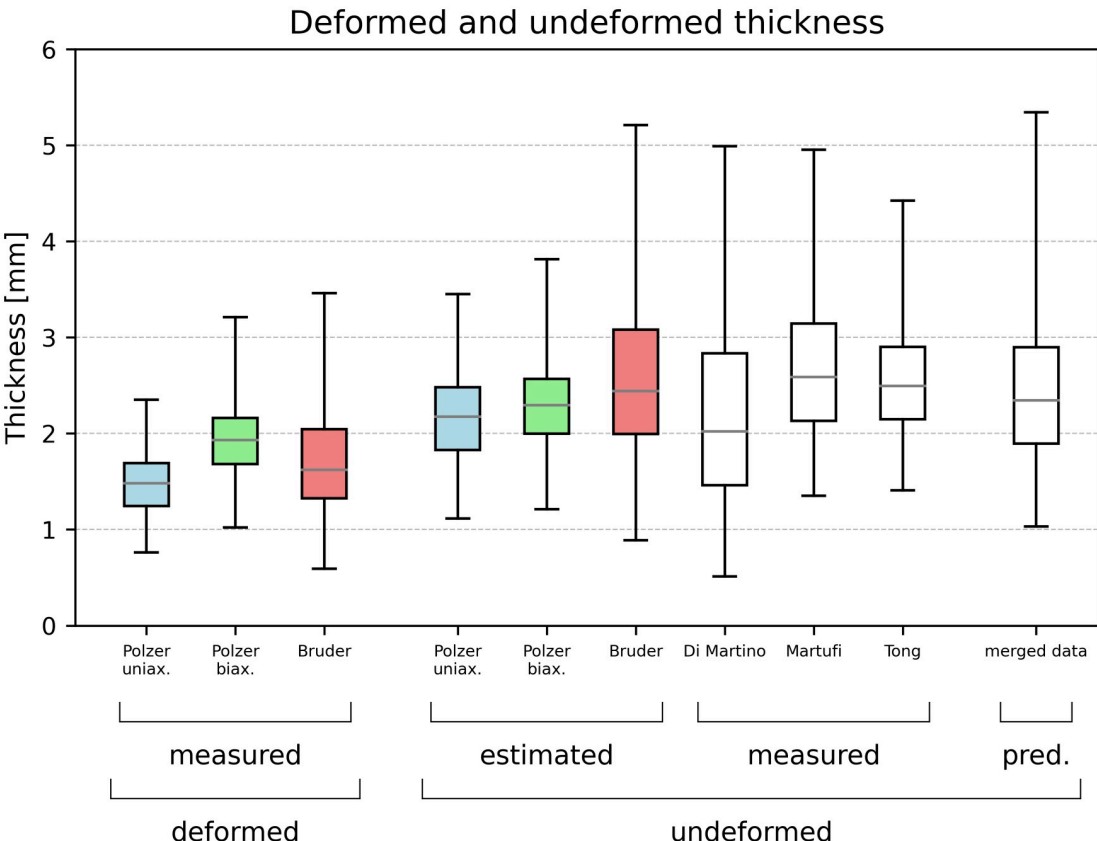

**Fig 6. Measured deformed thicknesses (boxplots 1–3), their transformations to undeformed thicknesses (4–6), measured undeformed thicknesses (7–9), and the boxplot of a virtual data set from the predictive distribution.** Whiskers mark data set extremes, while boxes denote the 1st and 3rd quartiles.

constructed using the theoretical quantiles of lognormal distributions with parameters obtained as point estimates from the sample means and standard deviations by the Method of Moments. For comparison, the last boxplot [10] illustrates the predictive distribution from the proposed model. This distribution predicts undeformed wall thickness utilizing all the available data. More specifically, the estimates of 98 theoretical quantiles of the predictive distribution were used to produce the boxplot, which corresponds to the average size of the data sets. The range (max-min) of means/medians of the 6 original data sets is decreased from 1.22/1.11 mm to 0.55/0.57 mm (i.e., reduced by 55/48%) when the undeformed thickness is considered (see Table 1).

## 3.4 Posterior predictive distribution

The posterior predictive distribution predicts the undeformed wall thickness of a new sample based on information extracted from all six data sets. Fig 7 compares the normalized histogram of the undeformed thicknesses (the measured undeformed thicknesses combined with the estimated undeformed thickness of the samples measured as deformed ones) with a histogram of data sampled from the posterior predictive distribution. The posterior predictive distribution is not a distribution of a known functional form, nevertheless, it can be approximated by a lognormal distribution $LN (\mu = 0.85, \sigma = 0.32)$ for practical purposes. Fig 7 shows it can accurately approximate the predictive distribution even in the area of low wall thicknesses as shown in

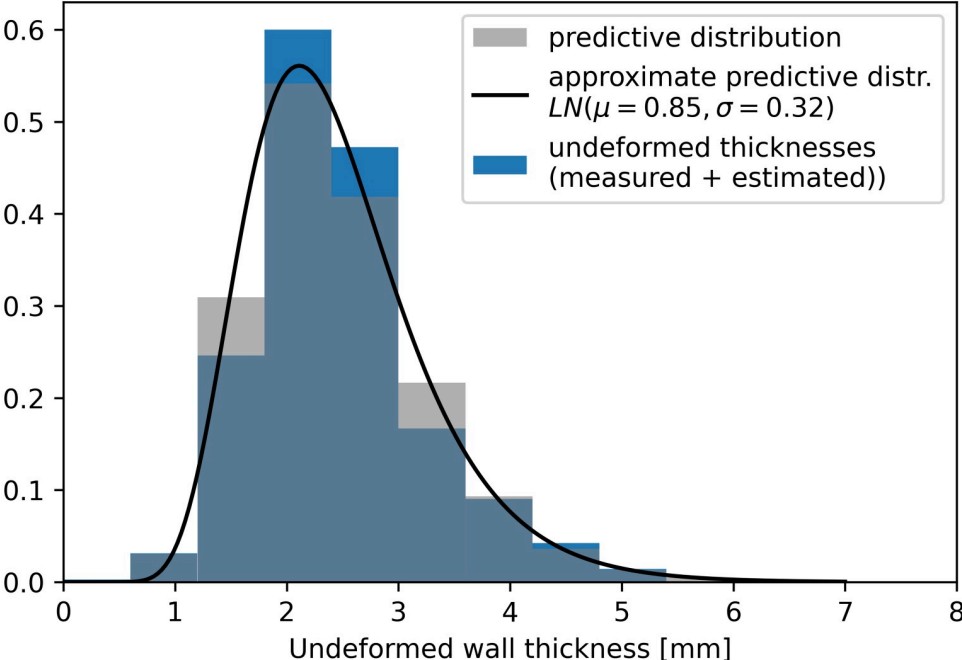

**Fig 7. Comparison of the normalized histogram of the undeformed thicknesses (blue), the posterior predictive distribution of undeformed wall thickness (grey), and its lognormal approximation (the black line).**

Table 2. The density of the lognormal approximation is also plotted in Fig 7 (the black line). Using the predictive distribution, we evaluate the mean/median of the undeformed wall thickness 2.46/2.34 mm and a 95% prediction interval [1.25, 4.39] mm. The Robustness analysis showed the predictive distribution of the undeformed thickness remain virtually unchanged regardless of the choice of prior distributions. For details, see S2 File.

## 4 Discussion

In this study, we have proposed a stochastic model based on Bayesian statistics to account for wall thickness data from various studies with different contact pressures. This is a critical step in the biomechanical rupture risk assessment of an AAA, where a credible distribution of unloaded wall thickness is crucial [8, 20]. Our analysis yielded a mean/median undeformed wall thickness of 2.46/2.34 mm and a 95% prediction interval [1.25, 4.39] mm. These values are close to the thickness $2.27 \pm 0.23$ $mm$ reported by Tong et al. [36] who also used a

**Table 2. The values of the predictive cumulative density function and its lognormal approximation for the AAA undeformed wall thicknesses below 1 mm confirm that the lognormal distribution is reasonably accurate.**

| AAA undeformed wall thickness $t_1$ [mm] | Probability of $t < t_1$ | |
|---|---|---|
| | Posterior predictive distribution [%] | Lognormal approximation [%] |
| 0.5 | <1.0E-7 | 7.1E-05 |
| 0.6 | 5.0E-04 | 1.1E-03 |
| 0.7 | 0.011 | 0.008 |
| 0.8 | 0.044 | 0.040 |
| 0.9 | 0.140 | 0.142 |
| 1.0 | 0.380 | 0.395 |

contactless thickness measurement. Since our results are based on data from various groups, we believe it to be the most accurate value available so far. Moreover, we also provide the parameters of the lognormal population distribution of undeformed wall thickness with parameters $\mu = 0.85$, $\sigma = 0.32$ so they can be used in any stochastic analysis for an AAA rupture risk assessment [8, 20]. These parameters are based on a total of 456 samples from 6 studies, so we can consider it the most robust prediction of an aneurysmal wall distribution available. A combination of data acquired by various groups helped us diminish the problem with an inaccurate description of the tails of the wall thickness distribution due to the low number of patients with a very low wall thickness (see Table 2). However, such data combination is not straightforward because various groups measure and report wall thickness differently, and that is why we had to propose the complex Bayesian stochastic model to overcome this obstacle.

Another novelty lies in the proposed sub-model, which allows for the consideration of data presented both as summary statistics and as complete data sets. Of course, considering data available only in the form of summary statistics is not ideal, and part of the information is inevitably lost, and estimation of its parameters are more or less fuzzy (see Fig 2). On the other hand, disregarding such data would also inflict information loss since they are based on a significant number of samples. Herein we provided a solution on how to incorporate such data. Exploiting all available information is crucial, especially in cases where hundreds of samples cannot be easily gathered, as is the case of less frequent diseases like thoracic aneurysms or aortic dissection, to name a few from the field of vascular pathology.

## 4.1 Contact pressure is a major source of inter-group variability

Our results demonstrate that accounting for contact pressure reduces the inter-group variability by about half (see Fig 6 and Table 1). This underlines the importance of accounting for an undeformed wall thickness. Nonetheless, other sources of inter-group variability, remain hidden. The presence of calcifications or other histological parameters (such as the density and waviness of collagen fibers or the ratio of inflammatory or adipose cells) certainly affects the sample radial stiffness and thus the undeformed thickness. Therefore, we advocate reporting information on a sample-specific basis so it can be referred to in future statistical models. Readers can find the sample-specific data we acquired in this study in the supplementary material.

A comparison of the estimated undeformed mean wall thickness to studies using caliper measurements, which result in significantly lower mean wall thickness of 1.57 mm [19], 1.5 mm [16], and 1.57 mm [17], further raises questions about whether a caliper is a suitable tool for soft tissue wall thickness measurement. It is important to mention that the analyzed tissue is extremely compliant radially, and a pressure of magnitude as small as 4 kPa generates a shift in the measured wall thickness of some 25% (see Fig 5). Additionally, calipers are optimized for measuring stiff metal-like materials and have very narrow contact surfaces; thus 4 kPa could be easily generated by barely noticeable force. An older study using an operator-independent resistivity-based tool reported a comparable mean thickness value of 2.09 mm [13] which is in very good agreement with our results considering that the study included 29 samples from only 5 patients.

## 4.2 Intra-patient variability is 30% lower than inter-patient variability

The analysis of the combined data shows that intra-patient variability cannot be neglected, although it is approximately 30% lower than inter-patient variability (see Fig 4). To illustrate the intra-patient variability, consider a typical patient whose parameter $\mu_i$ is equal to the posterior mean of $\mu_{pop}$. For $\sigma_{pat}$ equal to its posterior mean, the 0.025 and 0.975 quantiles of undeformed thickness are 1.6 a 3.3 mm, respectively.

However, we should note that the available data provide less information about inter-patient variability compared to intra-patient variability, as shown in Fig 4. Roughly speaking, a difference between two thickness values from a single patient can be explained only by intra-patient variability. On the other hand, a difference between two thickness values from two distinct patients can be explained by both inter-and intra-patient variability. This is the reason why the posterior density of $\sigma_{pop}$ is less concentrated compared to the posterior density of $\sigma_{pat}$. This highlights the necessity of analyzing the inter and intra-patient variability separately, as demonstrated herein on the aneurysmal samples for the first time. On the other hand, it should be emphasized that our intra-patient variability is obtained from data coming from the anterior part of the AAAs, only due to the unavailability of data acquired from other parts of the AAAs. This is because, in common practice, the tissue from other parts of AAA is not resected. There is a possibility to obtain data from other parts of AAA during autopsies as in [16, 19], however, those studies measured thickness via calipers so they could not be included.

### 4.3 An apparent mismatch between mechanical properties from various tests

Besides the contact pressure, the estimation of unloaded thickness $T$ also requires information about the mechanical properties of the sample. We initially used the mean response from the planar biaxial tests performed by Vande Geest et al. [29] and let the two constants responsible for initial stiffness $c$ and $k$ be predicted by our model based on data that provided some information about stiffness in a radial direction. For validation, we took the results, predicted equibiaxial response (in terms of stretches), computed the initial stiffnesses in both a circumferential and axial direction (see Table 3) and compared them with other studies [36–39]. It can be seen that our values are in good agreement with the data of Chuong and Fung [39] who tested non-aneurysmal arterial tissue in the radial direction (similarly as we did). Our results are also comparable with a subset of patients above 60 years old in the study [38] (which re-analyzed the data from [29]). Contrary to our results, the studies that performed planar biaxial tests report 6–8 times higher initial stiffness. The discrepancy might be attributed to the constitutive model behavior used as it is not clear to which extent can HGO, FUNG, or Four-fiber family constitutive models capture an arterial response in a radial direction when fitted with planar biaxial data since those data are not generally available. Another cause might lie in the incomplete information about the underlying wall thickness measurement setup as often the contact pressure is not reported [29, 38] and different studies report distinctly variable mean wall thickness. For instance [18] reports a thickness around 1.2 mm while [29, 38] reports 1.32 mm which is by some 45–50% lower than the undeformed thickness our model predicts

**Table 3. A comparison of the initial stiffness in a circumferential and axial direction predicted by our model in an equibiaxial tensile test with other studies not used for model construction.**

| Study | $E_0$ circ [kPa] | $E_0$ axial [kPa] | Tissue type | Type of sample testing |
|---|---|---|---|---|
| This study 2.5%/50%/97.5% Quantile | 4.8/21/84 | 2/10/42 | AAA | Radial compression + planar biaxial tension (data from vande Geest et al. [29]) |
| Tong et al. [36] | 115 | 119 | AAA | Planar biaxial tension |
| O´Leary et al [37]* | 156 | 127 | AAA | Planar biaxial tension |
| Chuong and Fung [39] | 21 | 21 | Rabit thoracic aorta | Radial compression |
| Ferruzzi et al [38] >60 years | 37 | 96 | AAA | Planar biaxial tension |
| Ferruzzi et al [38] <30 years | 148 | 132 | AAA | Planar biaxial tension |

*The mean response was corrected for variable wall thickness and a known contact pressure reported elsewhere(18).

based on a wider range of data. A third cause can be a persisting lack of data due to which we had to use a relatively strong assumption that the mechanical properties are the same across the whole population. That is, strictly speaking, in contradiction with reality, and this deficiency in the model can be addressed once more data on radial compression in AAAs are available.

### 4.4 Limitations

Several limitations of this study could be addressed in the future. Firstly, we assumed an ideal uniaxial compression loading state, which may not be entirely realistic. The studies [9, 20, 40] use a micrometer smaller than the sample, which makes the deformation somehow constrained by the surrounding tissue, while the setup used by our group [23] with a glass on a sample also potentially limits the deformation due to friction. This effect increases with increasing deformation which means only a minor effect is to be expected on the presented results as the deformation during sample measurements should be rather minor. Then, the intra-patient variability may be underestimated since there are no suitable data from other parts of the AAA but the anterior. Another limitation arises from laser measurements of the thicknesses which were considered to be true undeformed thicknesses, but they may suffer from an unclear surface definition of adventitia. The adventitia tissue is known to consist of loose wavy collagen fibers, so without its flattening, it is not clear which surface the laser detects (18). This may ultimately lead to overestimating the thickness in studies employing laser measurements. Another limitation is the presence of calcifications within the tissue which affects the resulting radial stiffness significantly. Currently, no group reported the stiffness for each individual case together with information about calcifications. It may partially explain the presence of very high undeformed thicknesses (above 4mm) in the data set [20]. Therefore, our model could wrongly evaluate the undeformed thickness if the deformed thickness was rather high and the sample was calcified. Occasionally, other data sets reported such high thickness as well (see Fig 6) meaning we could not exclude those data a priori. The model can be improved once the data sets with at least qualitative information about the role of calcifications in radial stiffness are published. On the other hand, we are mainly focused on the probability of an AAA having a low wall thickness which would be affected only negligibly.

## 5 Conclusions

In this work, we proposed an approach for combining available AAA wall thickness data into a single cohort while respecting the different contact pressures at which they were acquired. The resulting cohort can be described via a lognormal distribution with a median of 2.34 mm and parameters $\mu = 0.85$ and $\sigma = 0.32$ which can be used in an AAA rupture risk assessment. Next, we showed that the intra-patient variability is only 30% lower compared to inter-patient variability. The results show that the data from the included groups can be combined when accounting for different contact pressures and can be used in the stochastic rupture risk assessment of an AAA. Finally, we showed that the wall thickness should always be reported together with the contact pressure used for that measurement for better replicability.

## Supporting information

**S1 File. Sample specific data including patient-specific medical information and wall thickness.** This file includes all the gathered sample specific data about wall thickness used in this study.
(XLSX)

**S2 File. Robustness analysis using alternative prior distributions.** This file includes detailed description of the steps in the robustness analysis used to confirm independence of the Bayesian analysis used in this study.
(DOCX)

## Acknowledgments

We would like to thank Dr. Vojtěch Man for his help with the experimental thickness measurements.

## Author Contributions

**Conceptualization:** Luboš Kubíček, Robert Staffa, Stanislav Polzer.

**Data curation:** Luboš Kubíček, Robert Staffa.

**Formal analysis:** Jan Kracík, Luboš Kubíček.

**Investigation:** Jan Kracík, Stanislav Polzer.

**Methodology:** Jan Kracík, Stanislav Polzer.

**Project administration:** Luboš Kubíček.

**Resources:** Robert Staffa.

**Supervision:** Robert Staffa.

**Validation:** Jan Kracík, Stanislav Polzer.

**Visualization:** Stanislav Polzer.

**Writing – original draft:** Jan Kracík, Stanislav Polzer.

**Writing – review & editing:** Jan Kracík, Luboš Kubíček, Robert Staffa, Stanislav Polzer.

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
