## [Decision Letter · Decision Letter 0]

6 May 2024

PONE-D-24-02030Contact pressure explains half of the abdominal aortic aneurysms wall thickness inter-study variabilityPLOS ONE

Dear Dr. polzer,

Thank you for submitting your manuscript to PLOS ONE. After careful consideration, we feel that it has merit but does not fully meet PLOS ONE’s publication criteria as it currently stands. Therefore, we invite you to submit a revised version of the manuscript that addresses the points raised during the review process.

We look forward to receiving your revised manuscript.

Kind regards,

Eyüp Serhat Çalık

Academic Editor

PLOS ONE

Journal Requirements:

4. Please remove your figures from within your manuscript file, leaving only the individual TIFF/EPS image files, uploaded separately. These will be automatically included in the reviewers’ PDF.

Additional Editor Comments:

I congratulate the authors for working on this important topic. I believe that your manuscript will provide important data on the hemodynamics of abdominal aortic aneurysms. However, the reviewers have the following major concerns about the manuscript. Please revise your manuscript according to the suggestions and provide point-by-point answers. I wish you success.

Reviewers' comments:

Reviewer's Responses to Questions

**Comments to the Author**

1. Is the manuscript technically sound, and do the data support the conclusions?

Reviewer #1: Partly

Reviewer #2: Yes

2. Has the statistical analysis been performed appropriately and rigorously? 

Reviewer #1: I Don't Know

Reviewer #2: Yes

3. Have the authors made all data underlying the findings in their manuscript fully available?

Reviewer #1: Yes

Reviewer #2: Yes

4. Is the manuscript presented in an intelligible fashion and written in standard English?

Reviewer #1: No

Reviewer #2: Yes

5. Review Comments to the Author

**Reviewer #1: **First of all, I want to congratulate the authors for their work and for the manuscript.

Reviewer comments regarding formals and language

• The authors do not follow the submission guidelines (https://journals.plos.org/plosone/s/submission-guidelines) as they have the “Author summary” between the introduction and the abstract, they use abbreviations in the introduction that are not mandatory, and so on

• They do not have a uniform citation style, example: Introduction line 3 “(1)” versus line 5 “[2]”.

• Some typos remained within the manuscript, for example “gave written informed consent., and…”

• At some points I struggle to follow the language / the text, maybe it is due to the fact, that I am not a native speaker. Therefore, despite Grammarly has been used, I would think the manuscript could benefit from the consultation of professional language editing.

Reviewer comments regarding the manuscript

• Following the title, the authors state that “contact pressure explains half of the abdominal aortic aneurysms wall thickness inter-study variability.

• The used Baysian modelling and own, newly generated data on the aortic wall thickness, as well as already published data.

• The data set the authors provide contains several parameters (basic characteristics: patient number, sample number, sex, body height, body weight; descriptive parameters of the abdominal aortic aneurysm: suprarenal aortic lumen diameter, suprarenal aortic outer diameter, local ILT thickness, AAA diameter, calcification orientation; information on co-morbidities: hypertension, smoker, HLP, COPD, blood / pulse pressure).

• The basic problem is we know that there is a high inter- and intraindividual variance, for example demonstrated in the ascending aorta (different changes convexity versus concavity). Thus, we cannot assume that everywhere in the aneurysm the tissue has the same biomechanical properties.

• I was not able to find any information in the manuscript on the sampling site and how the authors controlled for “matching” sampling sites.

• Additionally no detailed histological data are presented. Differences in the exact degeneration can lead to different biomechanical properties, such as mucoid extracellular matrix accumulation +/- smooth muscle cell nuclei loss and so forth.

• The authors distinguish between presence and absence of “strong calcification”. But, they do neither define the parameter nor do they provide an adequate description of calcification. But we know that there is a big difference, especially in terms of biomechanis, intimal vs. focal medial vs. larger confluent calcifications and so forth.

• Taken together, despite the very sophisticated and well-described analyses, I do not think that neither the data nor the analyses allow for the strong conclusion the manuscript draws. Without a proper description of the architecture (i.e., histopathological assessment – are the samples comparable?) and ensuring comparable sampling sites such a conclusion is for my understanding not possible. Therefore, I would recommend a major revision as I think the work the group is doing is important and can help to “move the field” by helping to increase “power” of the data and usability of data of other studies!

**Reviewer #2:** The present study devised a Bayesian statistical method for predicting the patient-specific AAA wall thickness by utilizing clinical data on wall thickness associated with a variety of patient-specific conditions. This research represents a substantial advancement in the field of AAA mechanics. However, significant concerns remain that require proper consideration and resolution.

1. The statement “One of the most critical factors is wall thickness” is still remained to be a conjecture. Particularly, it showed that “One study found that ruptured AAAs have thicker walls(15) but others did not confirm that(16,17)” and “despite a known increase in wall strength in thinner samples(21–23)”, which is contradictory to the first statements. Thus, the first statement ought to be a verified hypothesis. This statement relies on the particular assumption that the mechanical properties are uniform and homogenous. It has been observed that the posterior wall of healthy aortas is thinner and stiffer, suggesting the possibility that a stress-mediated vascular adaptation process increases stiffness as a compensatory mechanism. Additionally, AAAs may be subjected to varying contact pressure at specific locations, such as pushing to the vertebral columns, where altered mechanical properties may result from the contact.

2. “To do this, the mentioned constitutive model was used to simultaneously fit all available biaxial responses of aneurysmal tissue published in (29) to obtain a mean response.” It is unclear why neither uniaxial nor biaxial experiments were conducted in the present study. It should be possible to validate whether this 3D model is comparable to the HGO model, given that the previous HGO models were based on only 2D measurements and it is uncertain whether the 3D HGO model results in having ‘unique’ sets of model parameters with all 3-D stretches. It is doubtful whether biaxial responses from references can be explicitly extrapolated from this radial test in the absence of validation.

3. All the AAA tissue samples were only obtained from the anterior side, and it can be a critical limitation for predicting overall thickness and rupture analysis.

4. Please clarify why very different contact pressure values (pc_uni vs. pc_biax) were used for the contact pressure tests.

5. It needs to define the deviatoric part in Eq. (2). Since B is FF^t, eigen values are squared from F. So why lambda_r is not squared?

5. “ referencing a declination of each family of fibers from the circumferential direction." Specify the unit (radian?).

6. Correct “ = (| −2⁄3 )”

7. Define “FPK”

6. PLOS authors have the option to publish the peer review history of their article (what does this mean?). If published, this will include your full peer review and any attached files.

Reviewer #1: No

Reviewer #2: **Yes: **Seungik Baek

---

## [Author Response · Author response to Decision Letter 0]

20 Jun 2024

Response to reviewers

The authors appreciate the valuable suggestions and comments of the reviewers, and the manuscript has been revised accordingly. Reviewers’ comments and our Point to point response is attached in separate file at the end of the generated pdf.

---

## [Decision Letter · Decision Letter 1]

19 Aug 2024

PONE-D-24-02030R1Contact pressure explains half of the abdominal aortic aneurysms wall thickness inter-study variabilityPLOS ONE

Dear Dr. polzer,

Thank you for submitting your manuscript to PLOS ONE. After careful consideration, we feel that it has merit but does not fully meet PLOS ONE’s publication criteria as it currently stands. Therefore, we invite you to submit a revised version of the manuscript that addresses the points raised during the review process.

We look forward to receiving your revised manuscript.

Kind regards,

Eyüp Serhat Çalık

Academic Editor

PLOS ONE

Additional Editor Comments:

Dear Authors

I must say that we are pleased that you have revised and uploaded your manuscript in a proper way. Your manuscript has been reviewed by three referees, one previous and two new. There were concerns especially about the statistical methodology and hence one of the reviewers was a biostatistician and their suggestions are below and I am confident that your manuscript will reach a higher level of quality in line with these suggestions. Please have the final version of your revised manuscript edited by a native English speaking expert. I wish you success.

Reviewers' comments:

Reviewer's Responses to Questions

**Comments to the Author**

1. If the authors have adequately addressed your comments raised in a previous round of review and you feel that this manuscript is now acceptable for publication, you may indicate that here to bypass the “Comments to the Author” section, enter your conflict of interest statement in the “Confidential to Editor” section, and submit your "Accept" recommendation.

Reviewer #1: All comments have been addressed

Reviewer #3: (No Response)

Reviewer #4: (No Response)

2. Is the manuscript technically sound, and do the data support the conclusions?

Reviewer #1: Yes

Reviewer #3: Yes

Reviewer #4: Yes

3. Has the statistical analysis been performed appropriately and rigorously? 

Reviewer #1: I Don't Know

Reviewer #3: I Don't Know

Reviewer #4: No

4. Have the authors made all data underlying the findings in their manuscript fully available?

Reviewer #1: Yes

Reviewer #3: Yes

Reviewer #4: Yes

5. Is the manuscript presented in an intelligible fashion and written in standard English?

Reviewer #1: Yes

Reviewer #3: No

Reviewer #4: Yes

6. Review Comments to the Author

Reviewer #1: Dear Stanislav Polzer and colleagues,

Thank you very much for the revision of the manuscript! I want to congratulate you for the work you put in the manuscript! And thank you for giving background information with your answers to the reviewer’s comments.

Just two very minor points, I personally think you can easily and swiftly include:

Point (A):

During the revision, you included the following sentence: “This is because it is not possible to extract tissue from other parts of the AAA without threatening the patient so the only possibility to obtain data from other parts is to harvest AAAs from autopsies (19,35).”

I know that this “thesis” is published in some instances. But there are conditions, in which resection even of the posterior parts are done, for example, in the rare mycotic aneurysm, where it can be (rarely) necessary to remove the infected tissue. Thus, the statement, that this is not possible, is for my understanding not correct. I would recommend the following: “This is because in the routine tissue from other parts of AAA is not resected.” With such a general statement you are “safe”, I personally think.

Point (B):

During the course of the abdominal aorta from the diaphragm to the bifurcation the aortic wall changes the “wall properties” in terms it becomes more and more a “muscular vessel”. Therefore, especially in long aneurysm, anterior does not equal anterior. In your answer to the reviewer comments you explained that you “knew roughly the position from which the tissue was harvested”. Could you please just provide this localization somewhere in the methods section, for example, samples of approximately at height of the renal arteries? In case I did not capture the information during the re-review, I apologize for this unnecessary comment!

After these two minor points are addressed, I think the manuscript should be ready to be accepted.

Thank you very much for your important work and looking forward to read your paper!

Best wishes!

Reviewer #3: This is an incredibly interesting paper, but unfortunately, the authors could not unlock its full potential mainly due to the drawbacks of the analyzed studies/datasets, which resulted in significant limitations such as possibly much higher inter- and intra-patient variability. Nevertheless, I would like to express my utmost respect to the authors, as this is one of those studies that raises an important questions and can spur further research. In addition, the authors provided detailed characteristics of each patient, which certainly can aid other teams in conducting similar studies.

1) When I was invited to review this manuscript, only the abstract was available to me. Given the nature of this study, I believe this paper needs to be reviewed by a biostatistician. I lack sufficient expertise to provide an opinion on some aspects of this paper, but I can and will review other parts that are more closely related to cardiovascular medicine. That being said, I have very little to add beyond what has already been mentioned by previous reviewers.

2) "professional language editing service" mentioned by the authors doesn't seem to be very professional, as there are still many errors in the manuscript, some of which are very basic. For example, "our results is based" (Discussion, 1st paragraph; incorrect noun-verb agreement), "vena cava inferior" (Introduction, 2nd paragraph; incorrect word order), etc. Apart from grammatical errors, many sentences just don't sound natural. Please, have a native speaker revise your paper.

3) I read the authors’ response to Reviewer 1 about the “strongly calcified” column in the supplementary file. I don’t think that the authors’ response is sufficient since virtually any calcification (not just “strong calcification”) fits their description. The description needs to be more elaborate and clear and must be provided in the supplementary file as well.

4) The authors need to be more elaborate and clear. Their responses to the reviewers are rather detailed and easy to understand, whereas the changes applied in the manuscript lack cohesion and appear much more confusing (e.g. the passage about calcifications), mostly due to poor English.

5) Any inclusion/exclusion criteria? What aneurysms were included in this study? Symptomatic? Asymptomatic? Ruptured?

6) Fig. 6 needs to be fixed. There are no spaces between words, making it difficult to read. Also, please upload figures in the correct order next time.

Reviewer #4: There are some comments about the equations and validation method.

In equation 1, the subscript k is not written (k1 and k2). On the other hand, these symbols are not fully explained.

The two symbols Ψiso and Ψaniso are undefined.

Recheck the definition of symbols in equation 1 in lines 113-115.

In equation 3, I think Wiso and Waniso should be replaced by Ψiso and Ψaniso. Also, in formula 2, the subscript λ is not correct.

In line 135, why is λ_r considered in the range of 0.61 to 1?

In Table 2, the scale of the first column (from the left) is not written correctly. Also, this table is not referred to in the text.

It is suggested to perform the sensitivity analysis for the robustness of the results compared to the selection of other non-informative and vague prior distributions.

In addition to the methods mentioned in Section 2.5, why have simulations or artificial data not been used to assess validation of the model?

Considering the almost big difference in the rupture risk of AAA in men and women, why the same parameter was considered for both?

Is it possible to investigate the effect of variables such as age and smoking, etc. on the rupture risk of AAA?

7. PLOS authors have the option to publish the peer review history of their article (what does this mean?). If published, this will include your full peer review and any attached files.

Reviewer #1: No

Reviewer #3: **Yes: **Bulat Abdrakhimov

Reviewer #4: **Yes: **Farzane Ahmadi

---

## [Author Response · Author response to Decision Letter 1]

21 Oct 2024

Our responses are attached in separate file

---

## [Decision Letter · Decision Letter 2]

11 Nov 2024

Contact pressure explains half of the abdominal aortic aneurysms wall thickness inter-study variability

PONE-D-24-02030R2

Dear Dr. Polzer,

We’re pleased to inform you that your manuscript has been judged scientifically suitable for publication and will be formally accepted for publication once it meets all outstanding technical requirements.

Kind regards,

Eyüp Serhat Çalık

Academic Editor

PLOS ONE

Additional Editor Comments (optional):

I would like to thank Dr. Stanislav Polzer and all the authors for their time and effort. I am pleased to announce that the manuscripts have been accepted for publication in this form. I wish you every success.

Reviewers' comments:

Reviewer's Responses to Questions

**Comments to the Author**

1. If the authors have adequately addressed your comments raised in a previous round of review and you feel that this manuscript is now acceptable for publication, you may indicate that here to bypass the “Comments to the Author” section, enter your conflict of interest statement in the “Confidential to Editor” section, and submit your "Accept" recommendation.

Reviewer #1: All comments have been addressed

Reviewer #3: All comments have been addressed

Reviewer #4: All comments have been addressed

2. Is the manuscript technically sound, and do the data support the conclusions?

Reviewer #1: Partly

Reviewer #3: Yes

Reviewer #4: Yes

3. Has the statistical analysis been performed appropriately and rigorously? 

Reviewer #1: I Don't Know

Reviewer #3: I Don't Know

Reviewer #4: Yes

4. Have the authors made all data underlying the findings in their manuscript fully available?

Reviewer #1: Yes

Reviewer #3: Yes

Reviewer #4: Yes

5. Is the manuscript presented in an intelligible fashion and written in standard English?

Reviewer #1: Yes

Reviewer #3: Yes

Reviewer #4: Yes

6. Review Comments to the Author

Reviewer #1: Dear Stanislav Polzer and colleagues,

thank you very much for revising the manuscript again. From my point of view, the manuscript is now acceptable!

Again, thank you for your efforts in the field of aortic biomechanics!

Best wishes!

Reviewer #3: Thank you for addressing all my comments and revising the manuscript. The text is now much more cohesive and clear. There are no major grammatical errors.

Reviewer #4: Dear authors,

Thank you for the revision of the manuscript.

Authors have answered all the comments and have done necessary correction in their manuscript.

7. PLOS authors have the option to publish the peer review history of their article (what does this mean?). If published, this will include your full peer review and any attached files.

Reviewer #1: No

Reviewer #3: **Yes: **Bulat Abdrakhimov

Reviewer #4: **Yes: **No

---

## [Editor Report · Acceptance letter]

19 Nov 2024

PONE-D-24-02030R2 

PLOS ONE

Dear Dr. Polzer, 

I'm pleased to inform you that your manuscript has been deemed suitable for publication in PLOS ONE. Congratulations! Your manuscript is now being handed over to our production team.

Kind regards, 

on behalf of

Dr. Eyüp Serhat Çalık 

Academic Editor

PLOS ONE